# Effect of Physiological Fluid on the Photothermal Properties of Gold Nanostructured

**DOI:** 10.3390/ijms24098339

**Published:** 2023-05-06

**Authors:** María Fernanda Amézaga González, Jazzely Acosta Bezada, Víctor Gómez Flores, Christian Chapa González, Jose Rurik Farias Mancilla, S. J. Castillo, Carlos Avila Orta, Perla E. García-Casillas

**Affiliations:** 1Insituto de Ingenieria y Tecnología, Universidad Autónoma de Ciudad Juárez, Av. del Charro no. 450 Nte. Col. Partido Romero, Ciudad Juárez 32310, CHIH, Mexico; al228153@alumnos.uacj.mx (M.F.A.G.); al229137@alumnos.uacj.mx (J.A.B.); victor.gomez@uacj.mx (V.G.F.); christian.chapa@uacj.mx (C.C.G.); rurik.farias@ciqa.edu.mx (J.R.F.M.); 2Departamento de Investigación en Física, Universidad de Sonora, Blvd. Luis Encinas y Rosales S/N, Hermosillo 83000, SON, Mexico; santos.castillo@unison.mx; 3Centro de Investigación en Química Aplicada, Blvd. Enrique Reyna Hermosillo No. 140, Saltillo 25294, COAH, Mexico; carlos.avila@ciqa.edu.mx

**Keywords:** gold nanoparticles, photothermal properties, light-to-heat efficiency

## Abstract

Colloidal gold particles have been extensively studied for their potential in hyperthermia treatment due to their ability to become excited in the presence of an external laser. However, their light-to-heat efficiency is affected by the physiologic environment. In this study, we aimed to evaluate the ability of gold sphere, rod, and star-shaped colloids to elevate the temperature of blood plasma and breast cancer-simulated fluid under laser stimulation. Additionally, the dependence of optical properties and colloid stability of gold nanostructures with physiological medium, particle shape, and coating was determined. The light-to-heat efficiency of the gold particle is shape-dependent. The light-to-heat conversion efficiency of a star-shaped colloid is 36% higher than that of sphere-shaped colloids. However, the raised temperature of the surrounding medium is the lowest in the star-shaped colloid. When gold nanostructures are exited with a laser stimulation in a physiological fluid, the ions/cations attach to the surface of the gold particles, resulting in colloidal instability, which limits electron oscillation and diminishes the energy generated by the plasmonic excitation. Fluorescein (Fl) and polyethylene glycol (PEG) attached to gold spheres enhances their colloidal stability and light-to-heat efficiency; post-treatment, they remand their optical properties.

## 1. Introduction

Colloidal gold nanoparticles (AuNPs) are among the most extensively studied nanostructures for biomedical applications. Due to their excellent chemical stability, bonding abilities, and optical properties, AuNPs have shown promise in treating epilepsy, syphilis, tuberculosis, rheumatoid arthritis, and cancer treatment [1,2,3]. These particles have been synthesized using different methods, but the Turkevich method is the most commonly used one due to its simplicity and versatility. This redox method allows the control of particle size, which is crucial in studying the effectiveness of treatment on physicochemical and biological properties [4]. There has been extensive research on surface modification of nanoparticles using polymers or other molecules to reduce toxicity, improve cell interaction, and increase residence time inside the human body to enhance drug delivery [5,6,7]. Core/shell nanoparticles have been used for controlled drug release. In a recent study, Liang et al. (2022) used gold–SiO_2_ core/shell nanoparticles loaded with hyaluronic acid for photothermal therapy and chemotherapy, improving cancer inhibition rate [8]. Lemire et al. demonstrated that attaching fluorescent molecules to nanoparticles could facilitate the identification of cancerous tissue [9]. The binding of AuNPs with molecules capable of undergoing a specific chemical interaction with tissue has improved drug release. The antibody–antigen reaction causes a specific and localized release of the drug to occur [10,11,12]. ErbB2-targeted AuNPs have been evaluated in a cell culture model, increasing the sensitivity and specificity of breast cancer cells [13]. Due to their intrinsic properties, these nanoparticles have been used in hyperthermia treatments. Hyperthermia is a cancer treatment that uses heat (>45 °C) to damage and kills tissue; however, specificity is necessary to avoid damaging healthy cells. Nanoparticles make this procedure more efficient because they can target or enter a cell membrane, causing tumor destruction by heat generation and dissipation from the inside out [14]. AuNPs transform absorbed light energy into thermal energy in the presence of external laser stimulation, causing an increase in the temperature of a specific tissue.

Targeted AuNPs have been designed to improve the effectiveness of control temperature increases by reducing laser exposure time in specific tissue [15,16,17]. Photothermal excitation occurs when a laser hits the electrons of the nanoparticles, generating a surface plasmon or plasmon resonance. This property depends on the shape and size of the particles. Minor modifications of morphological properties produce a change at the nanoparticle surface, which affects the surface plasmon resonance, as it alters the boundary conditions in terms of polarizability [18]. Jiang determined that light-to-heat energy conversion efficiency increased as the size of the AuNPs decreased because resonance wavelength depends on particle shape and size [19].

Shape modification of AuNPs substantially affects their colloidal stability, photothermal and biological properties. In biomedical applications, the light-to-heat conversion depends on how the morphological properties of AuNP colloids interact with biological entities. For example, rod-shaped AuNPs increase blood circulation time due to their large surface area and can remain in the blood for up to 48 h [20]. In this gold structure, the generated plasmon divides into two bands, which are positioned along an axis due to the elongated particles [21]. Ma et al. (2010) predicted the optical properties of silver star-shaped nanoparticles using a mathematical model. They changed the angle and number of peaks to conclude that an increasing peak would result in less effective resonance wavelengths [22].

The intention is to modify optical properties to eliminate cancer cells while causing minimal heating to any healthy tissue. So, ample literature addresses the controversy of how particle size affects the photothermal properties of AuNPs, as well as a report about the efficiency of light-to-heat conversion of AuNPs. However, this parameter depends on morphological properties and how colloidal stability is affected by the interaction with biological entities.

One of the limitations of colloidal gold nanostructures in biomedical applications is their instability in contact with physiological fluids. Various factors, including the high concentration of salts in physiological media, pH changes, and protein adsorption, can cause this instability. The colloidal instability of AuNPs affects light-to-heat conversion and their successful applications in biomedicine.

Previous research evaluated the photothermal properties of gold particles, but in biomedical applications, these nanoparticles must be in the form of colloids to achieve greater stability and biocompatibility. Therefore, the colloidal medium can affect the conversion of light to heat. The physiological medium contains varying concentrations of ions/cations, which can alter the stability and, consequently, the conversion of light to heat. The challenge of the AuNPs colloidal is to adapt to the phycological environments. For example, unlike cancerous tissue, healthy tissue manifests different blood flow, oxygen supply, nutrients, and pH [23]. As a result, the conversion and transfer of energy resulting from the plasmon resonance of gold nanoparticles can be affected by the different physiological conditions present in the body.

For this reason, the Turkevich method was used to synthesize AuNP colloids, modifying the shape of the gold nanoparticles. Sphere, rod, and star-shaped nanoparticles were used to study how colloidal stability and light-to-heat conversion efficiency are affected in different physiological fluids. We also evaluated the effect of fluorescein-polyethylene glycol coating on the photothermal properties of sphere-shaped colloids as a potential strategy for improvement and understood the effect of coating particles on light-to-heat conversion.

## 2. Results

### 2.1. Raw Gold Colloids

The raw gold colloids are materials obtained from the Turkevich method without any modifications. The sphere-shaped raw colloid has a pH = 6. Figure 1A shows the TEM image of an asphere-shaped colloid with an average particle size of 14 ± 3.6 nm, and Figure 1B depicts their principal diffraction planes related to the XRD pattern of nanoparticles. In Figure 1C, the rod-shaped gold colloid has an average length of 37 ± 6 nm and an average width size of 12 ± 2.8 nm; their corresponding XRD pattern with reflected planes can be seen in Figure 1D.

Figure 1E shows a TEM image of a star-like structure with an average particle size of 73 ± 21 nm. The stars have 5 to 8 peaks and different vertex angle peaks. Their electron diffraction pattern is shown in Figure 1F. Finally, the complex of AuNPs with the fluorophore and the polyethylene glycol (Spheres-Fl-PEG) have a particle size of 469 nm ± 183 nm according to SEM images (Figure 2).

Figure 3 shows the UV-Vis spectrum of gold nanostructures. The sphere-shaped gold colloid presents a broad band with a maximum absorbance of 520 nm. Concerning rod-shaped, two maximum absorbance bands at 590 and 710 nm are visible. In the star-shaped colloid, a slight increase in absorption is noticeable at 492 nm. When the gold spheres are conjugated with a fluorophore (Fl) and polyethylene glycol (PEG), Figure 3B shows a prominent absorption band appearing at 490 nm, and the peak corresponding to gold plasmon decreases. FI band absorption is at 490 nm.

The TSI values of gold colloids are shown in Table 1. The star-shaped raw colloid has a higher TSI value than the sphere and rod-shaped raw colloid. However, the lowest TSI value was obtained when Fluorescein (Fl) and polyethylene glycol were attached to spheres. When these raw colloids are in contact with a physiological fluid (pH = 7.4, blood plasma and pH = 7.2, breast cancer), all TSI values increase compared to the original. Sphere-shaped colloid has the highest TSI value at pH = 7.4 (blood plasma), while the rods, stars, and sphere-Fl-PEG at pH = 7.2.

Due to the crystalline nature of a nanoparticle system, we assume that our AuNPs, in the varying shapes obtained, exhibit a direct band gap, so we used the Tauc method to calculate their optical band gap [24]. Diffuse–reflectance measurements allowed us to calculate nanostructure band gap values (E_g_). For spheres, the E_g_ value was 2.16 eV, as shown in Figure 4A. In the case of the rod shape, the E_g_ value was 2.1 eV, as shown in Figure 4B, and a lower value of 1.89 eV was registered for the star-shaped AuNPs, see Figure 4C. E_g_ value of fluorescein is 2.44 eV, and when spheres were coated with Fi-PEG, their band gap value increased slightly to 2.35 eV (Figure 4E).

For the rod-type nanoparticles, $E_{g}^{R1}=1.46\;eV,$ corresponds to a wavelength of 849 nm, and $E_{g}^{R2}=1.46\;eV$ corresponds to 667 nm. However, significant signals or absorption edges are absent in the absorption spectrum. See Figure 2A and Figure 3B, plotted in purple. It can be concluded that, in the case of spheres, rods, and stars, the size of crystals diminishes as the band gap increases.

### 2.2. Photothermal Properties

#### 2.2.1. *Particle Shape Effect*

Figure 5 shows the temperature behavior of raw gold colloids in contact with a green laser. During the first 20 min, the sphere-shaped colloid exhibited a higher temperature than the rod-shaped and star-shaped colloids. However, after 25 min, sphere- and rod-shaped particles have the same temperature value. Finally, at 30 min, the rods (37 °C) and spheres (36 °C) reached the highest temperature, while the stars only reached 27 °C. The heat rate of sphere-shaped particles has a higher value of 0.7 °C/min, followed by rod-shaped at 0.44 °C/min and star-shaped particles at 0.36 °C/min.

#### 2.2.2. Effect of Attached Organic Molecules

When fluorescein-polyethylene glycol (PEG) is attached to spheres (Figure 6), no significant difference is observed in the temperature reached or the heating rate.

#### 2.2.3. Effect of the Physiological Environment

Figure 6 shows the photothermal properties when raw gold colloids are in contact with a physiological fluid with pH 7.4 (blood plasma) and pH 7.2 (breast cancer) exposed to laser light. In the case of sphere-shaped particles (Figure 7A), the maximum temperature reached in contact with simulated physiological fluids is 33 °C (pH = 7.4) and 32 °C (pH = 7.2), which is lower than the temperature reached by the raw colloid (36 °C). Concerning rod-shaped (Figure 7B), the temperature of the raw colloid reached (37 °C) decreased due to the presence of the physiological fluid to 33 °C and 32 °C at pH = 7.2 and pH = 7.4, respectively. Star-shaped colloid has similar behavior; the temperature decreased in contact with simulated fluids for the star-shaped nanostructures (Figure 6C). Finally, no significant change in temperature exists when Fl-PEG is attached to the sphere-shaped nanostructures.

Moreover, it is also possible to estimate the light-to-heat conversion efficiency of gold nanostructure in contact with electromagnetic wavelength (the green-laser; 532 nm) from a cited and presented model at point 4.4 of the methodology section in this work. Star-shaped have the highest light-to-heat efficiency than spheres and rod-shaped colloids. Using Fl-PEG coating on the sphere-shaped colloid increased efficiency by 35% compared to spheres without a coating layer.

Table 2 shows the compilation of the calculated efficiencies for the different shapes of AuNPs synthesized in this research paper.

### 2.3. Photodegradation

Thomas and Col. (2018) define photodegradation as nanostructure changes due to heat generated by laser irradiation [25]. Therefore, the optical properties of colloids were assessed after exposure to radiation and a physiological medium. Colloidal post-treatment is shown in Figure 8. The absorbance band of spheres, rods, and stars exhibits extremely low intensity, almost disappearing (Figure 8A–C). When spheres were coated with Fi-PEG (Figure 8D), absorbance intensity decreased less than it did for uncoated spheres (Figure 8A).

## 3. Discussion

We used the Turkevich method to obtain a sphere-shaped gold colloid [4]. The reduction of chloroauric acid is accomplished by rapidly adding sodium citrate at a temperature between 80 and 100 °C. Nucleation and growth of gold nanospheres are very rapid at 100 °C, resulting in an average particle size of 14.9 nm with a narrow distribution (S.D. = 1.9 nm) and a high surface area due to the small particle. During nucleation, the formation of the AuCl/SADC complex occurs due to a rapid exchange between the citrate anion and the Cl^−^ ion, forming an intermediate complex [Au (Cl_3_(C_6_H_5_O_7_)^−2^]. Sodium citrate acts as a reducing agent, resulting in acid pH = 6 due to the ions generated from the hydrolysis of AuCl^4^ [26]. This mechanism produces a sphere-shaped raw colloid with moderate stability (TSI = 30.6). Fluorescein (Fl) and polyethylene glycol (PEG) were employed to modify the intramolecular forces to enhance colloidal stability, resulting in the lowest TSI value of 2.94.

Modifying particle shape is carried out in two stages. HAuCl_4_ is reduced with NaBH_4_ in the presence of CTAB, forming a gold seeds solution with a particle size of approximately 2 nm. During the second stage, the anisotropic growth of the gold seeds is achieved by binding Au ions to the cationic micelles of the surfactant. Ascorbic acid reduces Au^+3^ attached to CTAB micelles to Au^+^, forming a stronger complex [27,28]. This complex interacts with the silver ions to form a thin layer deposited in the (110) plane, which restricts growth in this direction (100), resulting in rod-shaped structures with 32.3 ± 5.2 nm in length and 11.8 ± 2.8 nm in width. This rod-shaped raw colloid has a lower TSI value than a sphere-shaped colloid, indicating higher colloidal stability; this is attributed to free CTAB in the solution, which is considered an excellent stabilizer. However, it has a pH of 2.

We used a similar procedure to the rod method to obtain the star-shaped colloid. The gold seed solution was homogenized with AgNO_3_ and ascorbic acid. Ascorbic acid reduces HAuCl_4_ and AgNO_3_ simultaneously, and Au^+^ and Ag^+^ ions deposit themselves on the surface of gold seeds, enabling anisotropic nucleation. Ascorbic acid acts as a stabilizer and induces anisotropic growth of gold atoms, forming the points of a star that grows faster, resulting in a star particle size of 73 nm ± 21 nm [29,30]. This star-shaped raw colloid has similar stability (TSI value of 36.85) to spheres (TSI = 30.6) at a pH of 6, although the particle size is almost five times larger. However, the star peaks provide a high surface area, increasing contact with surfactant and colloidal stabilizers, resulting in similar intramolecular forces that stabilize the particles.

The raw gold colloids obtained directly from the Turkevich method have good stability, but their highly acidic pH renders them unsuitable for biomedical applications. In addition, when in contact with physiological fluids, which typically have a pH of 7.4, the acid nature of the colloids will lead to colloidal instability.

One of the properties that allow the use of these particles in hyperthermia treatments is their optical properties. When the AuNPs are exposed to a light beam, the electric field induces polarization of the conduction electrons, creating an electronic oscillation and a difference in charge on the nanoparticle surface. This acts as a restoring force, leading to a plasmon resonance effect that causes strong absorption and scattering of light at a specific wavelength [31,32,33]. The most widely used shape in biomedical applications is spherical. The maximum absorbance of sphere-shaped AuNPs was observed at 520 nm, a value similar to that reported in the literature. Kimling et al. (2006) determined that the maximum absorbance (LSPR) of gold nanoparticles depends on particle size and that this absorbance peak shifts when the diameter of the particle increases [34].

Modifying the photothermal properties of gold through morphological changes can make photothermal therapy more efficient. Asymmetric growth, such as rod-shaped particles, can result in drastic changes in the position of the peak. Free electrons oscillate in two different directions: transverse and longitudinal. Due to the change in polarization caused by geometric deviations, the maximum absorption peak splits into two bands. Anisotropic nanoparticles, such as rods, have multiple crystalline surfaces. It causes the surface plasmon polarizations to oscillate in two directions parallel to the longitudinal (long) axis. Therefore, the first absorbance band at 590 nm is due to the transverse axis, while the second band at 710 nm is due to the longitudinal plane of the structure. As a result, gold rods emit multiple plasmon modes consisting of two dipoles in opposite directions. In contrast, symmetrical nanospheres exhibit a single resonant plasmon due to their dipole shape [28].

The optical properties of gold star nanoparticles depend on various factors, such as their size, number, and vertex angle of the peaks. The sphere-shaped and rod-shaped colloids exhibit a narrow particle size distribution. In contrast, star-shaped and sphere-Fl-PEG colloids have a broader particle size distribution, which impacts their optical properties. Particle absorbance increases with larger particle size, so a broad particle distribution in the colloids results in a convoluted absorbance band. Ma (2010) found that the resonant wavelength of silver stars shifted towards high-energy blue light as the number of peaks increased. In our case, the gold star nanoparticles shifted up to 210 nm. The optical properties of AuNPs are influenced by the vertex angle peaks, with smaller angles exhibiting a larger resonant wavelength [21]. The resonance plasmon wavelength of the stars relates to the hybridization of the peaks and their larger surface area. The stars’ wide particle size distribution, variable peak numbers (5 to 8), and various vertex angle peaks give rise to a complex band. The absorption increases near 500 nm, similar to what was reported by Ramsey et al. [30].

The light emitted by gold nanostructures depends on the band gap energy. Variation in the band gap of the different nanostructures behaved as described by Kayanuma, where a short particle size has a more significant band gap value due to quantum confinement [35]. Spherical particles are the minor structures (14.9 nm) with an E_g_ value of 2.1 eV, whereas stars represent the largest particle (43.9 nm), so they have the smallest E_g_ value (1.89 eV). Coated spheres with Fl-PEG have a higher band gap value than uncoated particles because the coating acts as a stabilizer, as reported by J. Thomas [25]. The energy of the band gap required to excite a photon is inversely proportional to the maximum wavelength present in the UV-Vis plot, as shown in Table 2.

Hyepthermia is one application where optical properties can be used. Gold nanostructures exposed to a laser (532 nm, 300 mW, 30 min) absorb energy, which is transferred to the surrounding medium through plasmonic excitation (light-to-heat conversion). According to D’Acunto (2021), the transformation of light-to-heat of AuNPs takes place in about one ps, so this energy dispersal into the medium is responsible for the temperature increase, and this energy can be used to eliminate a specific body tissue using thermotherapy [36]. Furthermore, the modulation of this energy is necessary to destroy malignant cells without affecting healthy tissue, as these cells require only half of the energy of healthy cells [37]. The transferred energy depends on the physiological medium’s shape, size, and composition. The raw star-shaped colloid reaches a maximum temperature of 27 °C in 30 min, which is insufficient to damage the tissue. However, star-shaped and rod-shaped colloids raise a temperature close to 40 °C in 30 min used in hyperthermia treatments [21]. Spheres have the fastest heating rate due to their greater homogeneity. However, after 30 min, the rod-shaped raw colloid reaches a temperature slightly higher than that of spheres due to its anisotropy, as shown in Figure 5. In the rod-shaped colloid, the plasmonic oscillation occurs in two directions, increasing these from the visible to near IR region. On the other hand, the star-shaped raw colloid reaches the lowest temperature due to its non-homogeneous peaks (peak number and vertex angle). When Fl-PEG adhered to gold spheres, particle size increased from 13.9 nm to 463 nm. The polymer acts as an energy-transfer barrier, slowing the heating rate. However, the temperature reached by the physiological fluid is very similar to the uncoated spheres. In the complex, gold nanoparticles are the photothermal material; the presence of organic molecules does not affect light-to-heat conversion because the gold content is similar.

When raw gold colloids come in contact with physiological fluids, they can experience instability due to changes in pH. Therefore, it is essential to study the behavior of these colloids in physiological conditions to understand their stability and potential limitations and can guide the development of strategies to improve their performance for biomedical applications. The particles’ shape can interact differently with the physiological fluids due to their surface properties. The star-shaped nanoparticles have a larger size (43.9 nm) than the spherical ones (13.9 nm), which results in a smaller contact surface with the physiological fluid. Therefore, the particle/fluid relationship is diminished for stars compared to spheres. Although star-shaped have the highest light-to-heat conversion efficiency, they transfer less energy to the physiological fluid, resulting in a lower temperature. When AuNPs are excited, the energy generated is transferred to the surrounding medium through the heat convection mechanism, and the physiological fluid absorbs this energy. However, the structure’s size and the amount of gold in each colloid depend on the specific formation mechanism. As a result, the temperature changes in physiological fluid vary depending on the size and number of particles present. The body has distinct physiological fluids with different concentrations and pH, so the light-to-heat conversion and stability of the gold colloid are modified. According to Vaupel, cancer tissues have a lower pH than normal tissues; the average value of mammary carcinoma is 7.2, although it can range from pH 6.5 to 7.6.

In contrast, blood plasma has an average pH = 7.4, mainly due to the difference in ionic and cationic concentration [23]. As a result, more energy is required to raise the temperature of the blood plasma compared to a neutral fluid. Slight changes in pH, such as in blood plasma and cancer fluid, cause colloidal destabilization, affecting thermal energy transfer and raising the temperature, as shown in Figure 7. For example, when sphere-shaped and star-shaped raw colloids come into contact with a physiological fluid, they reach lower temperatures due to the colloidal destabilizing (higher TSI values meaning less colloidal stabilization, Table 1). Under laser light stimulation, ions/cations content in the physiological fluid are electrostatically attracted to the surface of gold nanoparticles, changing the electric charge balance between the Van der Waals attraction forces and the strong repulsions caused by the polymeric layer. Although stars exhibit higher light-to-heat efficiency, the morphology and surrounding fluid affect energy transferred to the medium, decreasing the temperature reached. In addition, the intramolecular force changes cause instability and aggregation of gold nanoparticles, diminishing the local surface plasmon resonance.

Coating spheres result in a 27% increase in their light-to-heat conversion efficiency compared to the uncoated nanoparticles because they have the highest colloidal stability even in contact with physiological fluids. The rod-shaped colloid in contact with physiological fluid shows an inverse behavior to sphere and star-colloid due to less instability.

According to Figure 8, the photothermal properties of gold nanostructures (rods, spheres, stars) were lost post-treatment. Photodegradation is irreversible, and the attached ions/cations to the surfaces of nanoparticles limit the electronic oscillation, which deteriorates the photothermal effect [37,38]. Additionally, instability weakens the colloidal properties, causing aggregation and segregation, affecting the optical properties. However, when a polymeric-coated shell is used, photodegradation is reduced, which is advantageous for their use in biomedical applications.

## 4. Materials and Methods

### 4.1. Synthesis of Raw Colloid

We synthesized spherical AuNPs using the Turkevich method with certain modifications [4]. First, to prepare the gold seed solution, chloroauric acid (HAuCl_4_, 2 mL 0.5 mM) was added to sodium citrate (Na_3_C_6_H_5_O_7_, 2 mL, 0.4 M) to obtain a reddish-gold solution. Next, the seed solution was maintained at 80 °C under constant agitation for 5 min to allow particle growth. The reaction was complete when the color of the solution changed to ruby red. Then, the colloid was cooled to room temperature and stored for later use. The material obtained directly from the Turkevich method is referred to as raw colloids.

### 4.2. Particle Shape Modification

The rod-shaped raw colloid was undertaken following a procedure described by Kimling [34], which involves the nucleation of gold seeds and their subsequent asymmetric growth. First, hexadecyltrimethylammonium bromide (CTAB) (C_18_H_42_BrN, 9 mL, 0.1 M), chloroauric acid (HAuCl_4_, 1 mL, 0.5 mM), and sodium borohydride (NaBH_4_, 1.2 mL 0.01 M) were mixed by constant stirring until a homogeneous seed solution was obtained. Next, an asymmetric growth solution was prepared by mixing CTAB (C_18_H_42_BrN, 18 mL, 0.1 M), chloroauric acid (HAuCl_4_, 2 mL, 0.5 mM), silver nitrate (AgNO_3_, 140 µL, 0.1 M), and ascorbic acid (C_6_H_8_O_6_, 140 µL, 0.1 M) to obtain a white-clear solution. Subsequently, 30 µL of the seed solution was added and stirred at 30 °C until a purple solution resulted. Finally, CTAB excess was removed by five consecutive washes with distilled water, followed by centrifuging at 8000 rpm for 15 min.

The procedure for using star morphology to obtain nanoparticles is as follows. First, chloroauric acid (HAuCl4, 10 µL, 0.5 mM) and hydrochloric acid (HCl, 10 mL, 1 M) were added to the gold seed solution (100 µL) until a homogeneous mixture was obtained. Next, silver nitrate (AgNO_3_, 100 µL, 0.01 mM) and ascorbic acid (C_6_H_8_O_6_, 50 µL, 100 mM) were added. The reaction was complete when the ruby solution became transparent. Finally, the star-shaped colloid was centrifuged at 3000 rpm for 15 min.

### 4.3. Gold Nanoparticles with Fluorescein and Polyethylene Glycol Colloid

Chloroauric acid (HAuCl_4_, 0.5 mL, 0.5 mM) was heated to 80 °C while stirring constantly. Then sodium citrate (Na_3_C_6_H_5_O_7_, 0.5 mL, 0.4 M) and fluorescein disodium salt (C_20_H_15_Na_2_O_5_, 0.006 g) were added. Next, ammonium hydroxide (NH_4_OH, 70%, JT Baker) was added to bring the pH = 9, and 500 µL of polyethylene glycol (PEG 550) was added. The product (AuNPs-Fl-PEG) was centrifuged at 9000 rpm for 10 min, resuspended with distilled water, and stored at 4 °C.

The amount of gold used during the three syntheses was the same for all morphologies.

### 4.4. Determination of Photothermal Properties

A mixture was prepared with 1 mL of the gold colloid and 1 mL of physiological fluid. Physiological fluid with pH = 7.4 (blood plasma) and 7.2 (breast cancer) were prepared using PBS tablets (10 mM of phosphate ions, 0.137 M NaCl, and 0.027 M KCl from Fisher Bioreagent), following the methodology described by Rodriguez (2022) [39]. The mixture was placed inside an insulating box to prevent energy dissipation. Two circularly polarized lasers were used to simulate the gold nanostructures. The first laser, with a 632 nm source (Coherent He NE, red laser, 10 mW), did not result in excitation of the gold nanostructure as the fluid temperature did not increase. However, the second laser with a 532 nm source (Spectra-Physics, green laser, 300 mW) successfully exited the gold, leading to photon emission and increased temperature of the physiological fluid. The mixture was continuously stirred, and the temperature was measured directly using a type K thermocouple. The beam spot size used was 0.82 mm at a working distance of 15 cm.

Temperature increases were recorded as a function of time. The heating rate was determined by applying Equation (1).
(1)$$T(t)=T_{0}+A_{1}e^{-t/t_{1}}$$
where $T_{0}$ is the steady-state value, which corresponds to the starting temperature as time tends to infinity (value of the asymptote); $A_{1}$ is the amplitude term, which represents the inflection point when the function no longer grows linearly at time *t* = 0; and $t_{1}$ is the growth rate represented by $\frac{1}{t_{1}}$.
(2)$$\frac{dT(t)}{dt}=-\left(\frac{1}{t_{1}}\right)\left(T-T_{0}\right)$$

Similarly, the energy balance can be considered as the interaction between a green laser of 532 nm with N gold nanoparticles suspended within a mass $m_{W}$ of deionized water by applying the equilibrium equation:(3)$$\left[m_{W}C_{p,W}+\sum_{i=1}^{N}m_{i}C_{p,i}\right]\frac{dT}{dt}=Q_{I}-Q_{ext}$$
where $m_{i}$ and $C_{p,W}$, $C_{p,i}$ represent the nanoparticle masses and heat capacities of water and nanoparticles, respectively; $Q_{I}$ and $Q_{ext}$ are the received and external energies from the laser and thermally converted, respectively, see [40,41].
(4)$$Q_{I}=I\left(1-10^{-A_{l}}\right)h$$
(5)$$Q_{ext}=hS\left(T-T_{0}\right)$$
(6)$$T-T_{0}=T^{*}$$
(7)$$\frac{dT^{*}}{dt}=A-BT^{*}$$

Considering the negligible thermal contribution of nanoparticles compared to the suspension medium, we can drive the following:(8)$$A=\frac{I\left(1-10^{-A_{l}}\right)h}{M_{W}C_{p,W}}$$
(9)$$B=\frac{hS}{M_{W}C_{p,W}}$$

The homogeneous associated equation with Equation (9) helped us to evaluate the B coefficient, $\frac{dT^{*}}{dt}+BT^{*}=0$, so that $q=\frac{T^{*}}{T_{max}-T_{0}}$:(10)$$d\left(ln\theta \right)=-Bdt$$
plotting lnθ as a function of time in a steady-state condition.

Let us consider T**_ss_** as the system’s steady-state temperature at thermal equilibrium, where *t* → ∞. and $T_{ss}^{*}$ the value at which $\frac{dT_{ss}^{*}}{dt}=0$, applying these conditions can be arrived at (9):(11)$$\left(T_{ss}-T_{0}\right)=\frac{I\left(1-10^{-A_{l}}\right)h}{B\left(m_{W}C_{p,W}\right)}=\frac{I\left(1-10^{-A_{l}}\right)h}{B\left(\rho_{W}V_{W}C_{p,W}\right)}$$
where $M_{W}$ or $m_{W}$, is the nanoparticle mass, also considering their densities, volumes, and heat capacities. From Equation (1), efficiency becomes clear:(12)$$\eta =\frac{B\left(T_{ss}-T_{0}\right)\rho_{W}V_{W}C_{p,W}}{I\left(1-10^{-A_{\lambda }}\right)}=\frac{\left(T_{ss}-T_{0}\right)hS}{I\left(1-10^{-A_{\lambda }}\right)}$$
$$B\rho_{W}V_{W}C_{p,W}\equiv hS$$

Preliminary data, $\rho_{W}=1\frac{g}{{\;cm}^{3}}$; $V_{w}=0.9543{\;cm}^{3}$; $S=0.6361725{\;cm}^{2}$; $C_{p,W}=4.2\frac{J}{g}$.

Laser power; I=0.3 W.

### 4.5. Characterization

Band gap energy was obtained from a Tauc plot using the absorption plot obtained from ultraviolet-visible (UV-Vis) spectrophotometry (Nanodrop 2000). Particle morphology was obtained by scanning (JEOL 7000F) and transmission electron microscopy (Hitachi 7700). The J-image software was used to determine particle size distribution using SEM images. The crystalline structure of the gold nanostructures was studied using an X-ray detector adapted to the TEM. Coating adherence onto nanoparticles was assessed by detecting the functional groups using Fourier transform infrared spectroscopy (Nicolet 6700). The surface area was measured using dynamic light scattering (DLS). The Turbiscan stability index (TSI) was used to evaluate colloidal stability through a UV-Vis microplate spectrophotometer. The absorbance of 200 mL of colloid was measured at 0, 15, 30, 120, 720, and 1440 min. The TSI value was calculated using Equation (13).
(13)$$TSI=\sqrt{\frac{\sum_{i=1}^{n}{\left(x_{i}-x_{BS}\right)}^{2}}{n-1}}$$
where $X_{1}$ is the average backscattering for each minute of measurement, $X_{BS}$ is the average $X_{1}$, and *n* is the number of scans.

## 5. Conclusions

We determined the capacity of gold sphere-shaped, rod-shaped, and star-shaped colloids to increase the temperature of blood plasma and breast cancer-simulated fluid under laser stimulation. In addition, the effect of colloidal stability, particle shape, and coating surface was studied on photothermal properties.

The light-to-heat efficiency of the AuNPs is shape-dependent. Star- and rod-shaped colloids show higher efficiency than sphere-shaped colloids. Nevertheless, the energy transferred to the surrounding fluid is reduced due to colloid instability caused by the adhesion of ion/cation of the physiological fluid onto the surface of the particles, resulting in a lower temperature reached from the physiological environment.

The presence of ion/cation in blood plasma and breast cancer-simulated fluid attached to AuNPs surface can lead to degradation of their optical properties. This causes the aggregation of particles, electron oscillation reduction, and the energy generated by plasmonic excitation, which could limit biomedical applications.

The binding of Fl-PEG to gold spheres enhances their colloidal stability and light-to-heat efficiency. In addition, it prevents instability caused by physiological fluids, resulting in a higher increase in temperature in a physiological environment. As a result, colloidal stability can be improved, and the effect of attached ion/cation on gold surface nanoparticles can be reduced using different coatings or surfactants, which would be beneficial for using these colloids in hyperthermia or drug delivery.

## Figures and Tables

**Figure 1 ijms-24-08339-f001:**
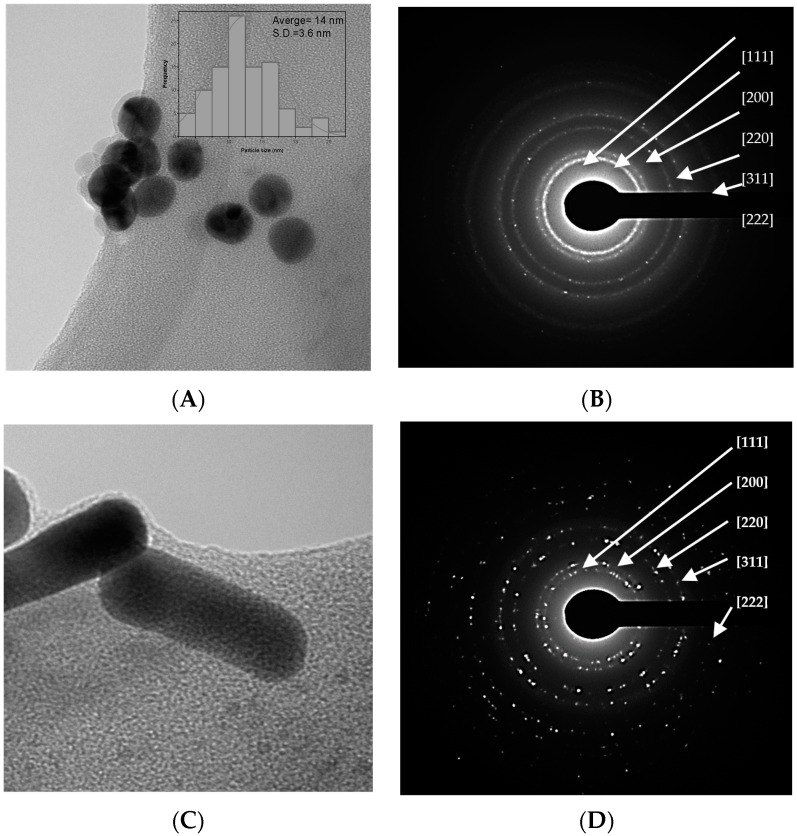
TEM images of (**A**) spheres, (**B**) XRD pattern of spheres, (**C**) rod-shaped, (**D**) XRD pattern of rods, (**E**) star-shaped gold colloids, and (**F**) XRD pattern of stars.

**Figure 2 ijms-24-08339-f002:**
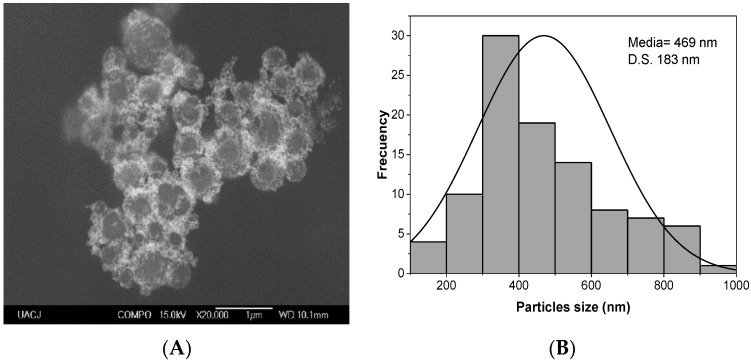
(**A**) SEM images and (**B**) particle size distribution of the complex spheres AuNPs with a fluorophore and the polyethylene glycol.

**Figure 3 ijms-24-08339-f003:**
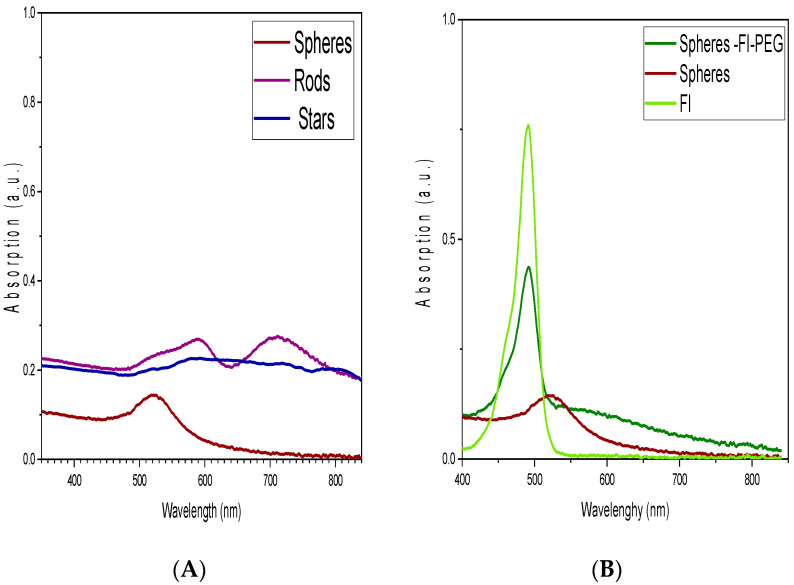
UV-Vis spectroscopy of (**A**) spheres, rods, stars, and (**B**) spheres and spheres-fluorophore-PEG (spheres-Fl-PEG).

**Figure 4 ijms-24-08339-f004:**
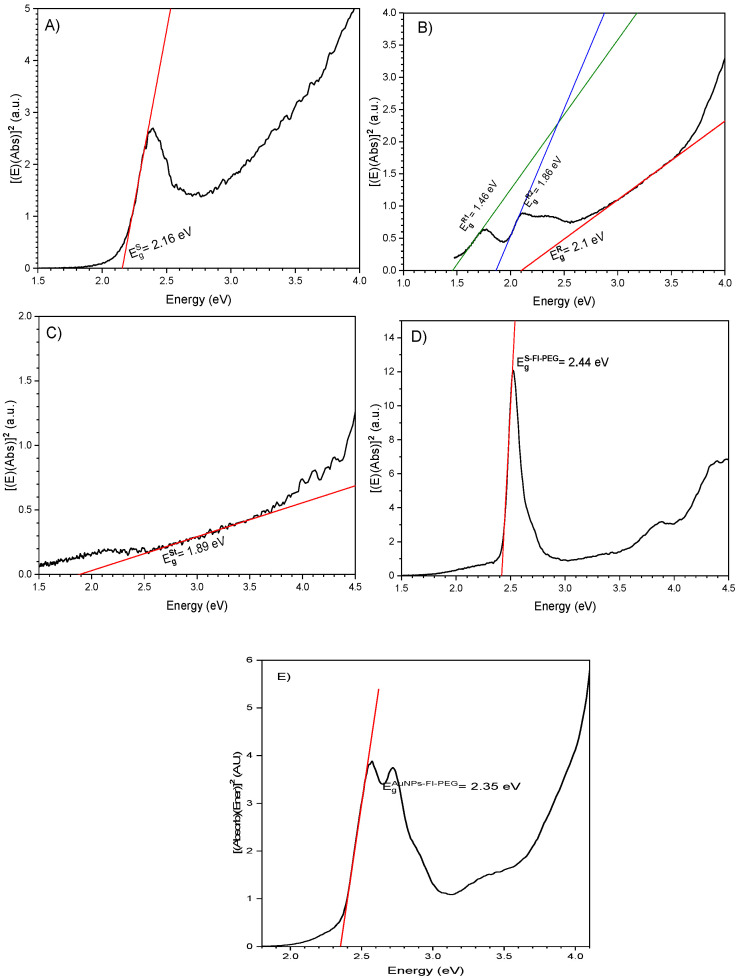
The optical band gap (E_g_) of (**A**) gold sphere-shaped, (**B**) rod-shaped, (**C**) star-shaped, (**D**) Fluorescein (Fl), and (**E**) spheres-FI-PEG.

**Figure 5 ijms-24-08339-f005:**
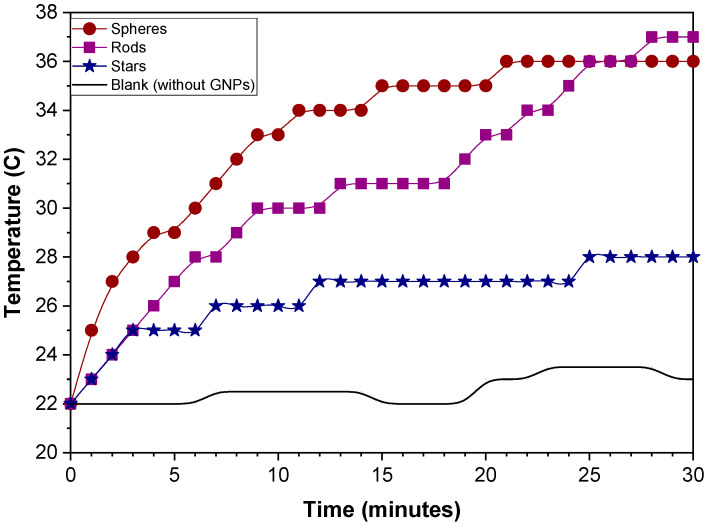
Temperature increases versus time plot for sphere-shaped, rod-shaped, and star-shaped raw colloids in the presence of green laser.

**Figure 6 ijms-24-08339-f006:**
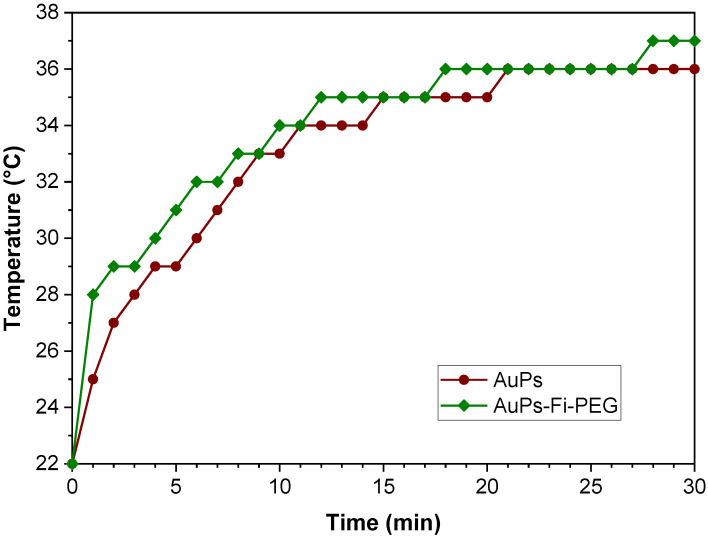
Temperature increases versus time plot for spheres and sphere-Fluorescein-PEG (Spheres-Fl-PEG) particles.

**Figure 7 ijms-24-08339-f007:**
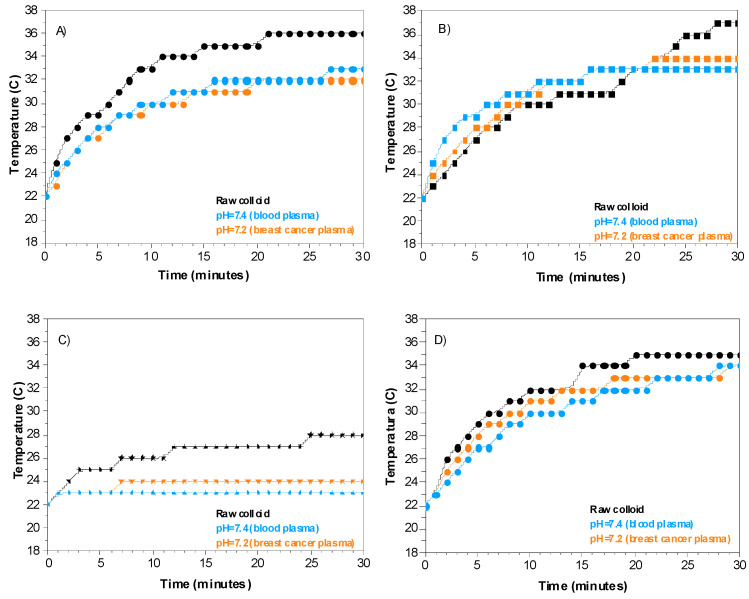
Temperature versus time plot for a different type of gold nanostructure in contact with a physiological fluid at pH = 7.4 (blood plasma) and pH = 7.2 (breast cancer) under green laser stimulation. (**A**) spheres, (**B**) rods, (**C**) stars, and (**D**) sphere-fluorescein-PEG.

**Figure 8 ijms-24-08339-f008:**
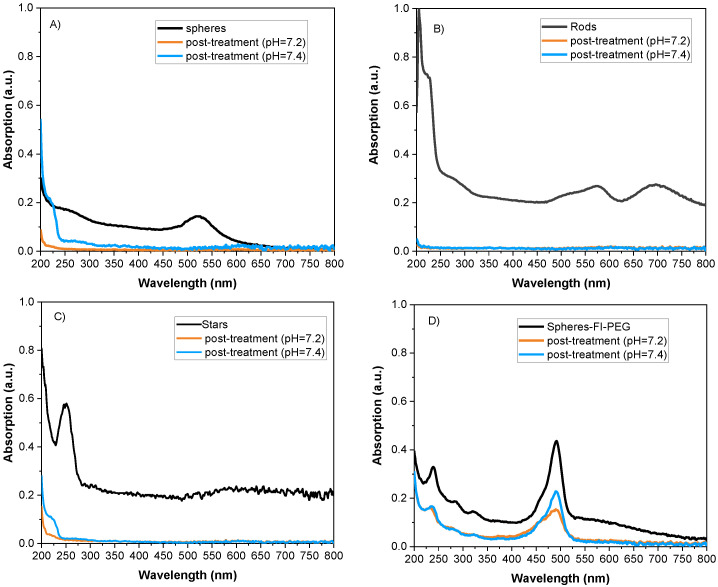
Absorbance post-photothermal treatment versus the wavelength of (**A**) sphere, (**B**) rods, (**C**) stars, and (**D**) sphere-fluorescein-PEG.

**Table 1 ijms-24-08339-t001:** Turbiscan stability index (TSI) of gold nanostructures.

Colloid	Turbiscan Stability Index (TSI)		
	Raw Colloid	pH = 7.2	pH = 7.4
Sphere-shaped	30.61	62.44	65.36
Sphere-Fl-PEG	2.94	4.38	3.26
Rod-shaped	14.71	41.99	30.52
Star-shaped	36.85	54.79	47.79

**Table 2 ijms-24-08339-t002:** Thermal efficiency conversion, η for different shape synthesized AuNPs.

NPs	T_ss_ (°C)	T_0_ (°C)	A_λ_ (532 nm)	$B\;\left(\frac{1}{s}\right)$	$hS(\frac{W}{{cm}^{2}\;{}^{\circ}C})$	η (%)
Spheres	36	22	0.2037	$5.84989\times 10^{-4}$	0.0023447	27.70
Spheres- FI-PEG	37	22	0.1189	$1.45\times 10^{-3}$	0.0020737	34.78
Rods	37	22	0.3928	$1.21\times 10^{-3}$	0.0048498	36.93
Stars	28	22	0.0537	$6.47092\times 10^{-4}$	0.0025936	37.77

## Data Availability

Data is contained within the article.

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
