# Peer review of "Effect of Physiological Fluid on the Photothermal Properties of Gold Nanostructured"

_ijms, 2023, doi:10.3390/ijms24098339_

Round 1

Reviewer 1 Report

This report (ID: ijms-2301486) is entitled “Effect of physiological fluid on the photothermal properties of gold nanostructured “ by research group Perla E. García-Casillas. This article is interesting and can make a good contribution to the scientific literature. However, the authors should present the application part. The reviewer suggests minor revisions before publication. Some specific comments on the manuscript:

1. ORIGINALITY:

This report results seem sufficiently novel but need to demonstrate photothermal application.

2. ABSTRACT:

The suggestion to the authors is to revise the abstract, summarize the problem, mention the importance of the methods used, and partly discuss results and conclusions.

3. METHODOLOGY:

*colloidal stability studies are needed, the effect of ionic strength and various types of electrolytes.

*Surface charge's role in photothermal properties can be examined to find a relationship with  photothermal properties

Explore performance enhancement strategy for improved photothermal activity

4. RESULTS AND DISCUSSION:

*Representation of results and discussion can be further enhanced and focus on scientific aspects.

*Specify limitations or challenges of materials to establish photothermal applications.

*Discuss the importance of surface chemistry, and show aggregation model correlate with photothermal properties

Correct spell mistakes, for example, Figure 7 Rew colloid

5. CONCLUSIONS:

Novelty, concluding remarks, and strategies for improved photothermal activity.  Discuss future implications  

Author Response

Your suggestions were of great contribution, attached result

Reviewer 2 Report

The paper deals out an investigate on the impact of the biological environment, shape, and coating on the photothermal properties of gold nanostructures. The study includes the light-to-heat conversion efficiency of star- and rod-shaped gold nanoparticles exceeded that of sphere-shaped nanoparticles. The paper is potentially intersting and the results could have a large impact on photothermal therapies. However, in the present form the paper lacks in presentation and the findings are not completely convincing. I suggest the authors the follwoing changes:

1)Please, use the acronym after intriduced first. For example, I suggest to use AuNp, instead of GNP to indicate the gold nanoparticles, in order to follow the acronym commnly used to denote such nanostructures.

2)  Figure 2B, the skewness of the particle size distribution must be discussed, and its impact on photothermal properties. This is also the case for all the particle size distribution plots presented in the paper.

3)  The section 4.4. should be introduced and presented just after the introduction, the authors should dedicate a section 2 to phothermal properties. It is no sense to mention equation 9 or 10 before to introduce equation 1 or 2.

3) Sections after References must be filled with the info related to the present paper.

4) The references are uncomplete. I suggest, among several paper, to cite D'Acunto et al. Nanotechnology, vol. 31, 192001, 2021.

5) Several sentences should be re-phrased. The English is generally correct,  but not always clear and fluid. Label figure7D should be translated in English. The general idea that the reader receives is that the paper was written with superficiality.

6) The authors should ad experimental work, for example, demonstratimg what happen with conditions different from: "The laser power used was 300 mW at a working distance of 15 cm", Are the author referring to standard protocols for power laser and distance? if we,, plesae specify.

Author Response

(The authors gave the same response as above.)

Reviewer 3 Report

The study of Gonzalez and colleagues describes the effects of gold nanoparticles with different shape, size, and surface chemistry on the physiological fluid such as pH. In addition, the various biological applications such as anticancer activity of the obtained gold nanoparticles was evaluated under irradiation. However, the effect of shape of gold nanoparticles were already reported that the spherical shape of gold nanoparticle was outstanding for the thermal energy transfer. Thus, the concentration of gold nanoparticle for the three sample (spherical, rod, and star) was not controlled as same particle number while the initial concentration of gold ion was same. And the chemical decoration such as FI-PEG was investigated only on the spherical gold nanoparticle. Therefore, the theses issues that need to be clarified to publish at this journal.

Author Response

Thanks for your suggestions, take it into account. I attach file

Reviewer 4 Report

In this manuscript by González et al., Effect of physiological fluid on the photothermal properties of gold nanostructured, the authors seek to evaluate the effect of pH (7.2 and 7.4) on the photothermal properties of gold structures with distinct morphologies.

This is a very interesting topic given the known versatile properties of gold nanostructures and their suitability for cancer treatment, among other clinical/biological applications.

However, I find that the manuscript is presented and written poorly. In my opinion, Results and Discussion could be together, since considerations made in the Discussion are very generic and established. Besides some data should be included for a better discussion of the data. For example, the Fl UV-Vis absorption spectrum is missing to correctly evaluate the optical bandgap; in Fig.5 a blank (w/o AuNP) should also be tested; besides the amount of gold in each sample should be controlled since it will affect the increase in temperature; there is not enough information regarding the set-up used to do the photothermal experiments of the so-called raw colloid, in what medium were these experiments made? Why have the authors chosen such a narrow pH gap to perform this study? It is known that the extracellular pH of breast cancer cells can be lower (6.9 – 7.1). Also, References should be updated, e.g.: in lines 74-76, literature examples showing the stated controversy should be provided. An extensive revision of the manuscript has to be made, some figures have non-English axis titles (fig. 6), or are not correctly mentioned in the text (fig.7).

Therefore, I cannot recommend the publication of the manuscript in its present form.

Author Response

(The authors gave the same response as above.)

Round 2

Reviewer 2 Report

In the revised version, the authors addressed all the issues. The new version of their paper can be published.

Reviewer 3 Report

Thank you for the sincere answers to the questions.

Reviewer 4 Report

The authors have answered the questions raised and changed the manuscript accordingly. Therefore, I can recommend its acceptance. Nonetheless, a thorough English revision is still needed.